# Affected cell types for hundreds of Mendelian diseases revealed by analysis of human and mouse single-cell data

Idan Hekselman[1], Assaf Vital[1], Maya Ziv-Agam[1†], Lior Kerber[1†], Ido Yairi[1], Esti Yeger-Lotem[1,2]*

[1]Department of Clinical Biochemistry and Pharmacology, Ben-Gurion University of the Negev, Be'er Sheva, Israel; [2]The National Institute for Biotechnology in the Negev, Ben-Gurion University of the Negev, Be'er Sheva, Israel

**Abstract** Mendelian diseases tend to manifest clinically in certain tissues, yet their affected cell types typically remain elusive. Single-cell expression studies showed that overexpression of disease-associated genes may point to the affected cell types. Here, we developed a method that infers disease-affected cell types from the preferential expression of disease-associated genes in cell types (PrEDiCT). We applied PrEDiCT to single-cell expression data of six human tissues, to infer the cell types affected in Mendelian diseases. Overall, we inferred the likely affected cell types for 328 diseases. We corroborated our findings by literature text-mining, expert validation, and reca-pitulation in mouse corresponding tissues. Based on these findings, we explored characteristics of disease-affected cell types, showed that diseases manifesting in multiple tissues tend to affect similar cell types, and highlighted cases where gene functions could be used to refine inference. Together, these findings expand the molecular understanding of disease mechanisms and cellular vulnerability.

*For correspondence:
estiyl@bgu.ac.il

†These authors contributed equally to this work

Competing interest: The authors declare that no competing interests exist.

## Editor's evaluation

The study presents analyses linking cell-types to monogenic disorders using over-expression of known disease-associated genes in single-cell data to identify disease-affected cell types for 328 Mendelian diseases. Overall, this important study combines multiple data analyses to quantify the connection between cell types and human disorders. Compelling analyses using stringent and rigorous statistical methodologies support the conclusions of this study.

## Introduction

Hereditary diseases affect 6% of the world population and typically lack cure. Identification of their genetic and molecular basis is challenging, limiting their diagnosis and treatment (*Ferreira, 2019*). Knowledge of tissues and cell types that manifest with pathophysiological changes in patients was shown to facilitate the understanding of disease mechanisms (*Hekselman and Yeger-Lotem, 2020*). For example, transcriptomic analysis of muscle tissues helped to genetically diagnose patients with rare muscle disorders (*Cummings et al., 2017*). Likewise, identification of a rare cell type affected by cystic fibrosis opened new avenues for treatment (*Montoro et al., 2018*; *Plasschaert et al., 2018*). However, whereas affected tissues may be evident for many Mendelian diseases, the exact cell types that manifest with pathophysiological changes within affected tissues are often unknown.

Owing to massive single-cell profiling of mammalian tissues (*Jones et al., 2022*), investigation of diseases in cellular contexts has become feasible. In particular, it was shown that genes whose

aberration leads to disease (i.e., disease genes) tend to be expressed preferentially in cell types that express pathology (denoted disease-affected cell types). For instance, 21/29 mouse homologs of human disease genes associated with nephrotic syndrome were shown to be upregulated in podocytes, the disease-relevant renal cell type (*Park et al., 2018*). In an inspiring manner, cell-type-specific expression of the cystic fibrosis gene CFTR in ionocytes of human and mouse airways revealed their role in cystic fibrosis (*Montoro et al., 2018*; *Plasschaert et al., 2018*). Likewise, the preferential expression of disease genes for 18 Mendelian muscle disorders revealed disease-affected muscle cell types (*Eraslan et al., 2022*). Thus, preferential expression of disease genes may serve as an indicator of disease-affected cell types, and has the potential to shed light on the cellular mechanisms that underlie Mendelian diseases.

Preferential expression of disease- and trait-associated genes has been used to illuminate cell types affected by complex traits in previously published studies (*Dai et al., 2021*; *Eraslan et al., 2022*; *Jagadeesh et al., 2021*; *Kim-Hellmuth et al., 2020*; *Rouhana et al., 2021*; *Zhang et al., 2022*). *Rouhana et al., 2021*, presented the ECLIPSER method, which tested whether the expression of genes mapped to trait-associated loci was enriched in specific cell types of a given tissue, when compared to a background set of genes associated with unrelated traits. The CSEA-DB repository applied a cell-type-specific enrichment analysis (CSEA): They assessed cell-type specificity of genes using t-statistics, and then tested whether the top 5% cell-type-specific genes were overrepresented in trait-associated genes (*Dai et al., 2021*). *Eraslan et al., 2022*, associated traits to relevant cell types by defining cell-type-specific gene modules and assessing their overlap with trait-associated genes. *Zhang et al., 2022*, introduced a polygenic single-cell disease relevance score (scDRS), which identified cells exhibiting excess expression of trait-associated genes. Interestingly, expression of trait-associated genes was weighted by trait-association and inversely by technical noise level in single-cell expression data, and the resulting polygenic scDRS score was compared to empirical distribution of control gene sets and all cells. *Jagadeesh et al., 2022*, presented sc-linker, which constructed continuous-valued gene sets that were differentially expressed in a cell type from healthy samples versus other cell types or versus the same cell type from disease samples, and then linked them to trait-related SNPs. Additional studies compared the differential expression of genes in cell types of diseased versus healthy tissues (*Mathys et al., 2019*; *Segerstolpe et al., 2016*) and showed that many trait-associated genes were cell-type specific (*Smillie et al., 2019*). *Guan et al., 2021*, defined cell-type specific gene interaction modules, which were overlapped with disease-associated genes. Lastly, the SC2disease database compared gene expression differences between cell types of diseased and healthy samples, among cell types of diseased samples, and among cell types of samples with varying degree of disease severity (*Zhao et al., 2021*). However, most efforts rarely validated or corroborated the inferred associations in an extensive manner.

Here, we developed a method that infers disease-affected cell types from the Preferential Expression of Disease genes in Cell Types (PrEDiCT). The summarized preferential expression of disease genes was compared to empirical distribution of control gene sets in the same cell type, allowing for the inference of associations between diseases and likely affected cell types. We applied PrEDiCT to 1,140 Mendelian diseases that manifest in six human tissues and inferred likely affected cell types for 328 diseases. We corroborated our findings by text-mining of PubMed records, expert curation, and by their recapitulation in mice. The resulting scheme and large-scale resource allowed us to show that diseases manifesting in multiple tissues tend to affect similar cell types, to refine inference of disease-affected cell types by focusing on gene functions, and to explore characteristics of disease-affected cell types.

## Results

### Preferential expression of disease genes indicates disease-affected cell types

To identify the cell types affected by Mendelian diseases with known disease genes, we developed the PrEDiCT scoring scheme (*Figure 1A*). Below we describe the PrEDiCT workflow, including data acquisition, PrEDiCT score calculation, and validation.

Data of annotated human single-cell transcriptomes were obtained from Tabula Sapiens (*Jones et al., 2022*). We focused on tissues with two or more samples with ≥800 sequenced cells that were

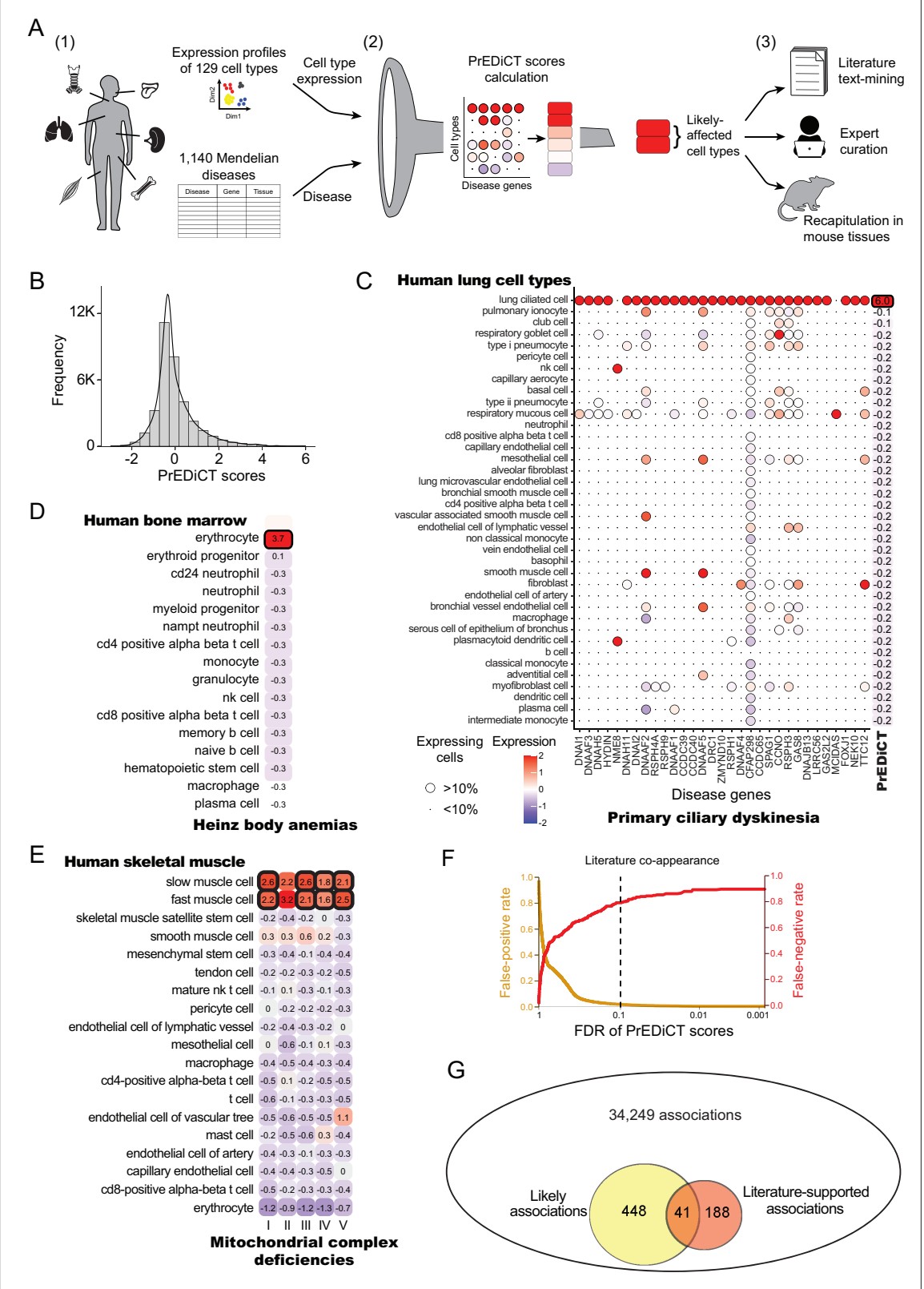

**Figure 1.** Overview of PrEDiCT calculation and assessment. (**A**) The PrEDiCT workflow. In step 1, we analyzed single-cell expression data from six human tissues and 129 cell types, and associated 1,140 Mendelian diseases with their affected tissues. In step 2, we calculated the preferential expression of disease genes in cell types of disease-affected tissues, used their median to produce the PrEDiCT score per disease and cell type, and assessed significance of each score. In step 3, we validated likely disease–cell-type associations (i.e., PrEDiCT ≥1, FDR <0.1) via literature text-mining, expert

*Figure 1 continued on next page*

*Figure 1 continued*

curation, and analysis of mouse single-cell expression data. (**B**) The distribution of PrEDiCT scores in human (median –0.25±0.93). (**C**) The preferential expression of genes causal for primary ciliary dyskinesia (PCD) and the PrEDiCT scores of PCD in lung cell types. Preferential expression values and the percentage of cells expressing a gene are indicated by the color and the size of each circle, respectively. The resulting PrEDiCT score is displayed on the right, colored by the score value. Bold outline marks likely disease-affected cell types. (**D**) PrEDiCT scores of Heinz body anemias across human bone marrow cell types depicted as described in panel C. (**E**) PrEDiCT scores of mitochondrial complex deficiencies across human skeletal muscle cell types depicted as described in panel C. Mitochondrial complex deficiencies were likely to affect slow and fast muscle cells, except for mitochondrial complex II deficiency whose PrEDiCT scores were highest yet insignificant in these cell types (FDR = 0.61 and 0.23, respectively). (**F**) False-positive (red) and false-negative (orange) rates of disease–cell-type associations (y-axis) across FDR thresholds (x-axis). Rates were estimated based on literature-supported pairs. The dashed line marks the FDR cutoff for likely associations. (**G**) The overlap between likely disease–cell-type associations (total of 489, left circle) and literature-supported associations (total of 229, right circle) out of all 34,249 possible associations. The overlap of 41 associations was significant (p<E-15, Fisher's exact test), supporting the validity of likely associations.

The online version of this article includes the following source data and figure supplement(s) for figure 1:

**Source data 1.** The Source Data file contains data used to generate *Figure 1B-G*.

**Figure supplement 1.** Disease-affected tissues.

**Figure supplement 1—source data 1.** The Source Data file contains the number of diseases with and without likely affected cell types per tissue.

**Figure supplement 2.** The PrEDiCT scheme.

**Figure supplement 3.** Distribution of diseases by the number of disease genes.

**Figure supplement 3—source data 1.** The Source Data file contains data of numbers of diseases per number of disease-assoociated genes with and without likely affected cell types.

**Figure supplement 4.** PrEDiCT scheme assessment using expert-curated associations.

**Figure supplement 4—source data 1.** The Source Data file contains data used to generate *Figure 1—figure supplement 4A, B*.

also sequenced in mice [(*Tabula Muris, 2018*); Methods]. Tissues included bone marrow, lung, skeletal muscle, spleen, tongue and trachea, and altogether were comprised of 129 cell types. Next, we associated these tissues with Mendelian diseases that affect them, based on their phenotypic abnormality annotations in Human Phenotype Ontology (HPO) database [(*Köhler et al., 2021*); Methods]. To assess the reliability of HPO annotations we compared them to manually curated annotations that were available in ODiseA database [(*Hekselman et al., 2022*); Methods]. This asserted 490 of 649 (76%) annotations available for these diseases in ODiseA, supporting the usage of HPO as an indicator of the disease-affected tissue. We then gathered the respective disease genes from the Online Mendelian Inheritance in Men (OMIM) database (*Amberger et al., 2019*). Overall, 1140 diseases and 2434 disease genes were associated to at least one affected tissue, with the majority associated to skeletal muscle (*Figure 1—figure supplement 1* and *Supplementary file 1*).

To calculate PrEDiCT scores, we computed the preferential expression of disease genes in each cell type relative to other cell types of the disease-affected tissue (Methods). Next, we set the PrEDiCT score of a disease in each cell type to the median preferential expression of the respective disease genes. Statistical significance of the score was determined using a permutation test (Methods; *Figure 1—figure supplement 2*). In total, we analyzed 34,249 disease–cell-type associations (*Supplementary file 2*). PrEDiCT score distribution is shown in *Figure 1B*.

We identified 489 (1.4%) 'likely' associations (PrEDiCT ≥ 1, FDR <0.1), covering 328/1140 diseases and 102/129 cell types. Although most likely associations involved diseases with a single disease gene, the fraction of diseases with likely associations was higher for diseases with multiple disease genes (*Figure 1—figure supplement 3*). As proof-of-concept, primary ciliary dyskinesia (PCD), which has 31 disease genes expressed in lung and is characterized by damaged ciliary machinery in lung ciliated cells (*Leigh et al., 2019*), was indeed likely associated with lung ciliated cells (*Figure 1C*). Additionally, Heinz body anemias, which is characterized by accumulation of inclusion bodies in erythrocytes (*Herman et al., 2023*), was indeed likely associated with bone marrow erythrocytes (*Figure 1D*). Lastly, compatible with the high demand for mitochondrial activity in muscle cells, four out of five mitochondrial complex deficiencies were likely associated with slow and fast muscle cells (*Figure 1E*). 'Mitochondrial complex II deficiency' also scored highest in slow and fast muscle cells yet its PrEDiCT scores were insignificant, potentially due to the stringency of our statistical analysis (*Figure 1E*). As a negative control, gracile bone dysplasia that might manifest with ankyloglossia (also known as 'tongue-tie') is not expected to inflict on cell types of tongue tissue, and indeed no likely

affected cell-type was detected (*Supplementary file 2*). Likewise, platelet-type bleeding disorder, which affects bone-marrow–derived platelets, had no likely affected cell-type in bone-marrow since platelets and their precursor cells (megakaryocytes) were missing from that tissue.

To assess at larger scale whether likely associations indicate disease-affected cell types, we turned to literature text-mining and to expert annotations. For text-mining, we postulated that diseases and their affected cell types will co-appear in the literature more frequently than expected by chance. To estimate co-appearance, we extracted PubMed records mentioning a disease or a cell type in our dataset and tested the significance of the co-appearance of all disease–cell-type pairs (Methods). Records were extracted by using Biopython package, which retrieves up to 9,999 records per term (*Cock et al., 2009*). We identified 229 'literature-supported' pairs that co-appeared in the literature more often than expected by chance (adjusted p<0.001, Chi-squared test; *Supplementary file 3*). PCD, for example, co-appeared significantly with lung ciliated cells, as well as with respiratory goblet, mucous, and basal cells (the latter are precursors of the other three). Based on literature-supported pairs, we estimated the false-positive and false-negative rates. The false-positive rates for likely associations were low (Methods; *Figure 1F*). Likely disease–cell-type associations were enriched for literature supported pairs (41/489, 8.4%) relative to all disease–cell-type associations (229/34,249, 0.7%; p<E-15, Fisher's exact test; *Figure 1G*). Lastly, we repeated the above analyses using expert-curated annotations of disease-affected cell types, showing a low false-positive rate (*Figure 1—figure supplement 4A*) and a higher enrichment for expert-curated associations (p=1.3E-4, Fisher's exact test; *Figure 1—figure supplement 4B*). Altogether, these results support likely associations as indicators of disease-affected cell types.

## Disease-affected cell types are recapitulated in mouse

To further assess whether PrEDiCT scores indicate disease-affected cell types, we tested whether matching cell types were likely affected in mice. For this, we downloaded mouse single-cell transcriptomes for the six tissues from *Tabula Muris, 2018*. These data consisted of 46 annotated cell types and some unannotated subsets (*Figure 2A*).

To improve the annotation of cell types, we reanalyzed the single-cell mouse transcriptomes (Methods). Altogether, we obtained 97 cell clusters across all six tissues (*Supplementary file 4*). Sixteen clusters overlapped considerably with cell types annotated by Tabula Muris and were thus annotated similarly. We annotated the 81 remaining clusters by careful examination of the expression of known cell-type marker genes (Methods), thereby improving cell type annotation per tissue (*Figure 2A*). For instance, the number of annotated cell types in skeletal muscle increased from six to 19. Newly annotated cell types included clinically relevant cell types, such as basement-membrane residing fibroblasts that are a main source of different collagens and are essential for skeletal muscle physiology (*Kivirikko et al., 1995*; *Zou et al., 2008*). Next, we aimed to identify similar cell types between human and mouse. This was based on expression of orthologous marker genes and the matchSCore2 package [Methods; (*Mereu et al., 2020*)]. As expected, a large variety of human cell types had a matching mouse cell type (82/129) and vice versa (70/97; *Supplementary file 5*). The impact of disease genes on the matching was negligible, as shown by repeating the identification of similar cell types upon excluding disease genes from the list of marker genes (Methods).

Next, we tested whether diseases affected matching cell types between human and mouse. For this, we calculated PrEDiCT scores in mouse cell types, based on expression of mouse orthologs of human disease genes. The distribution of PrEDiCT scores was similar between mouse and human, and included 380/24,638 (1.5%) likely disease–cell-type associations (PrEDiCT ≥ 1, FDR < 0.1; *Figure 2B*; *Supplementary file 6*). Compatible with our previous proof-of-concept cases in human, PCD likely affected mouse lung ciliated epithelial cells which match human lung ciliated cells (*Figure 2C*), and Heinz body anemias likely affected mouse bone marrow reticulocytes and erythroblasts which match human erythrocytes (*Figure 2D*). Likewise, four mitochondrial complex deficiencies likely affected mouse striated muscle cells, which matched human slow and fast muscle cells (*Figure 2E*). Similar to the results in human, 'mitochondrial complex II deficiency' scored highest in these cell types, yet the scores were insignificant (*Figure 2E*). We compared the PrEDiCT scores of all cell-type pairs of the corresponding tissues between the species. Whereas PrEDiCT scores of non-matching cell types did not correlate (r=−0.02, Spearman correlation), PrEDiCT scores of matching types were modestly correlated (r=0.38, p<E-15, Spearman correlation; *Figure 2F*). Of the 328 diseases with likely affected

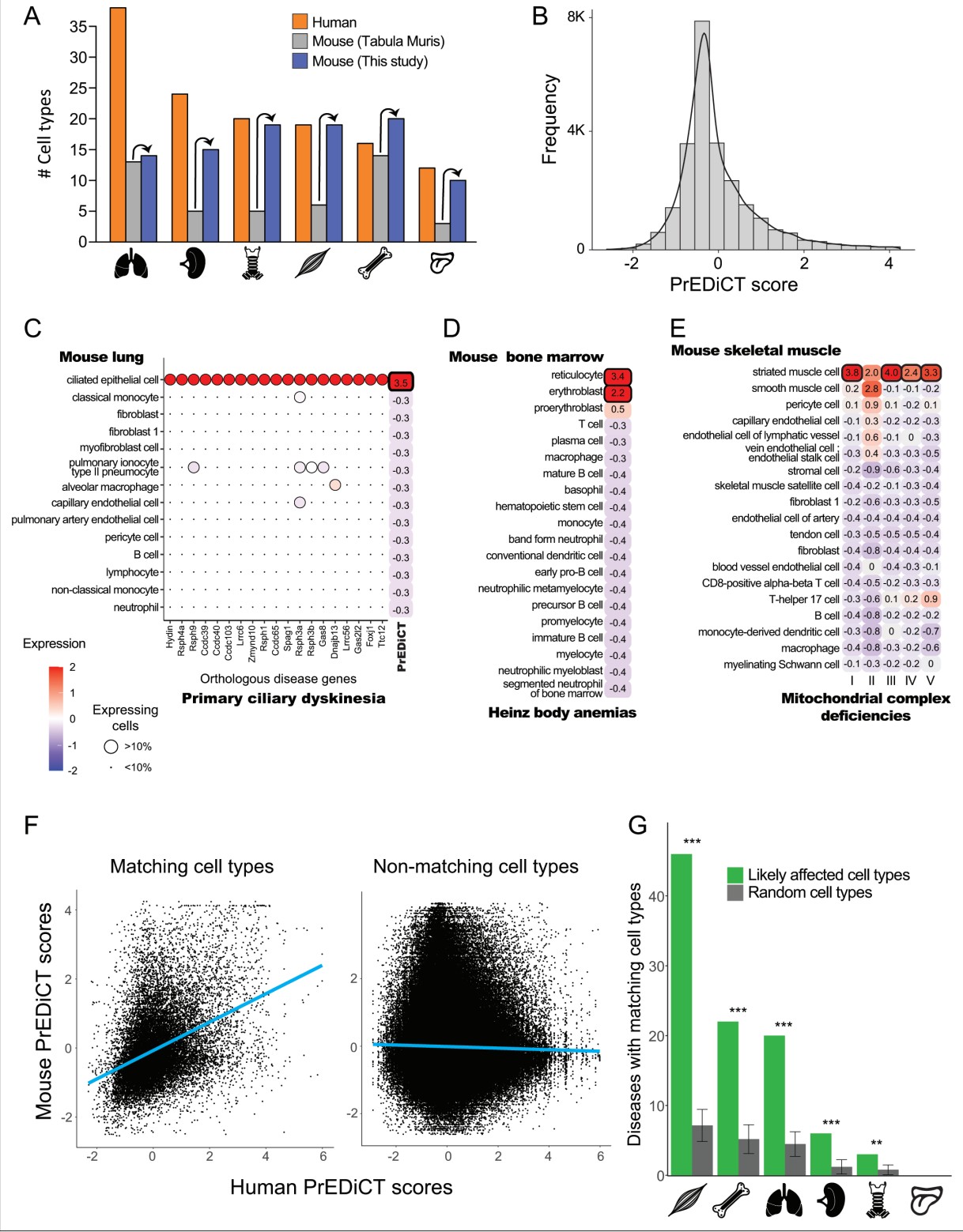

**Figure 2.** Recapitulation of disease-affected cell types in mouse. (**A**) The number of human cell types annotated by Tabula Sapiens [(***Jones et al., 2022***); red], and the number of mouse cell types annotated by [***Tabula Muris, 2018***; grey] and this study (blue). (**B**) The distribution of PrEDiCT scores in mouse. (**C**) The preferential expression of mouse orthologs of PCD disease genes and the PrEDiCT scores of PCD in mouse lung cell types. Preferential expression values and the percentage of cells expressing a gene are indicated by the color and the size of each circle, respectively. The resulting PrEDiCT score is indicated on the right colored by the score value. Bold outline marks likely disease-affected cell types. (**D**) PrEDiCT scores of Heinz

*Figure 2 continued on next page*

*Figure 2 continued*

body anemias across mouse bone marrow cell types depicted as described in panel C. (**E**) PrEDiCT scores of mitochondrial complex deficiencies across mouse skeletal muscle cell types depicted as described in panel C. Mitochondrial complex deficiencies were likely associated with striated muscle cells, except for mitochondrial complex II deficiency whose PrEDiCT scores were highest yet insignificant in these cell types (FDR = 0.65). (**F**) The correlation between PrEDiCT scores in human (X-axis) and mouse (Y-axis) cell types. Each dot represents a distinct pair. PrEDiCT scores of non-matching cell types did not correlate (left; *r*=−0.02, Spearman correlation), in contrast to PrEDiCT scores of matching cell types (right; *r*=0.38, p<E-15). (**G**) The cell types affected by the same disease in human and mouse tended to match each other (green) more than expected by chance (grey) according to 1,000 repeats in a permutation test. Error bars represent the standard deviation of the number of randomly matching cell types between the species. Adjusted **p<0.01 and ***p<0.001, permutation test.

The online version of this article includes the following source data and figure supplement(s) for figure 2:

**Source data 1.** The Source Data file contains data used to generate Figure 2A-G.

**Figure supplement 1.** The fraction of likely disease–cell-type associations in human that were recapitulated in mouse.

**Figure supplement 1—source data 1.** The Source Data file contains the numbers of diseases with a single, or multiple, disease-associated genes that were or were not recapitulated using mouse expression data.

cell types in humans, 97 diseases (30%) affected matching cell types in mice, a fraction that was larger than expected by chance (adjusted p<0.01, permutation test; *Figure 2G* and *Supplementary file 6*; Methods). This enrichment, and the observation that commonly affected cell types were not biased toward specific cell types, support the validity and generality of the PrEDiCT scheme.

## Diseases with multiple inflicted tissues affect similar cell types

Most diseases with a likely affected cell type manifested clinically in two or more tissues, and were denoted multi-tissue diseases (168/328, 51%; *Figure 3A*). We suspected that these diseases affect a cell type that is similar between the disease-manifesting tissues. To test this hypothesis, we identified matching cell types between tissues, and then examined whether cell types affected by the same disease were enriched for matching cell types. Matching cell types were identified using matchSCore2 package [Methods; (*Mereu et al., 2020*)]. Overall, we identified 840 pairs of matching cell types between tissues, of which 52% were immunocytes, including macrophages, neutrophils, T cells, B cells, as well as endothelia and fibroblasts. Next, we examined whether the cell types likely affected by the same disease were enriched for matching cell types. We found that 18% (30/168) of the diseases likely affected at least one pair of matching cell types, a fraction higher than expected by chance (adjusted p<0.01, permutation test; *Figure 3B* and *Figure 3—figure supplement 1*; Methods). For instance, chronic granulomatous disease (CGD) results in splenomegaly and pneumonia due to impaired phagocytes (i.e., neutrophils, macrophages, and monocytes; *Anjani et al., 2020*; *Leiding and Holland, 1993*). These cell types were indeed the likely affected cell types for CGD in both spleen and lung (*Figure 3C*). Also, CGD likely affected bone-marrow phagocytes, consistent with bone-marrow transplant being a curative treatment for this disease (*Leiding and Holland, 1993*).

## Cellular context prediction refined using gene functions

So far, we assumed that the genes underlying a disease work through a single cellular context. However, the same disease phenotype might arise from mechanisms with distinct cellular contexts. For example, the bone-marrow disease hyper-IgM immunodeficiency could arise from mutations in CD40 gene, affecting B cells, or mutations in CD40 ligand (CD40LG), affecting CD4 T cells (*Yazdani et al., 2019*). In such cases the PrEDiCT scheme might fail to infer the affected cell types. Indeed, only CD4 T cells were predicted as likely affected by hyper-IgM immunodeficiency (PrEDiCT = 1.6, FDR <0.048). We hypothesized that the cellular context of such 'multi-cellular diseases' could be revealed by separately applying the PrEDiCT scheme to subsets of disease genes with distinct functions.

We first focused on diseases that alter intercellular communication. We identified seven diseases in our dataset that were caused by mutations in genes encoding ligands and their receptors. Next, we applied PrEDiCT scheme separately to ligands and to receptors (Methods). As expected, hyper-IgM immunodeficiency was found to likely affect naïve B cells through CD40 (receptor; PrEDiCT = 2.9, FDR = 0.06), and CD4αβ T cells through CD40LG (ligand; PrEDiCT = 3.7, FDR <0.001; *Figure 4A*). Another example is autosomal recessive limb-girdle muscular dystrophy, which is caused by mutations in laminin-α2 (LAMA2) or its receptor α-dystroglycan (DAG1). By applying PrEDiCT scheme without subdividing the disease genes, we could not infer any likely affected cell type. Yet, by separately applying

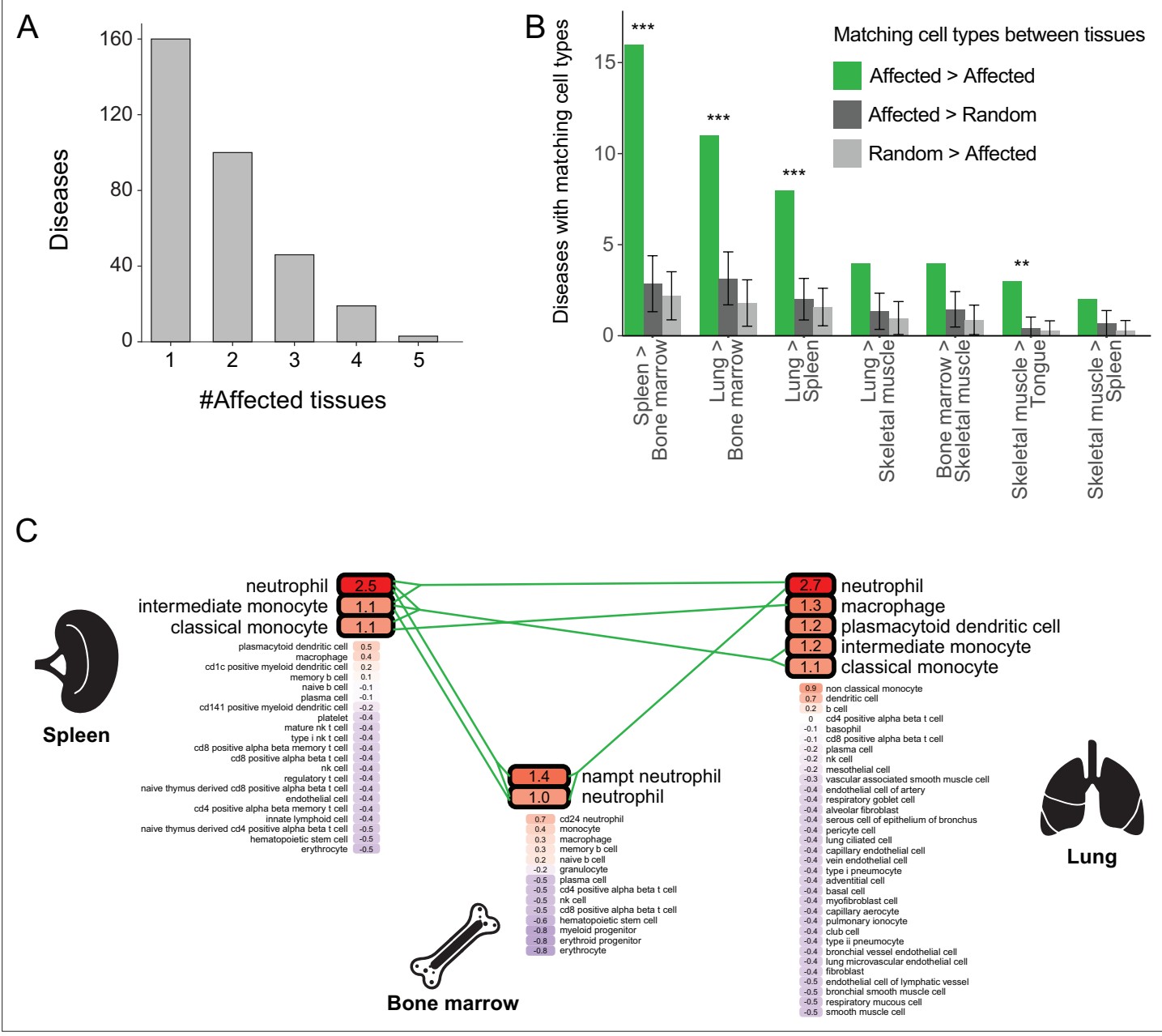

**Figure 3.** Multi-tissue diseases tend to affect similar cell types in those tissues. (**A**) The numbers of diseases with likely affected cell type across tissues (Y-axis) grouped by the number of affected tissues (X-axis). (**B**) The number of diseases that likely affect at least one pair of matching cell types between tissues (green). This number was higher than expected by chance (dark and light grey correspond to selecting cell types at random from the first tissue or the second one, respectively) according to 1,000 repeats in a permutation test. Only pairs of tissues with ≥2 shared diseases are shown. Error bars represent the standard deviation of the number of randomly matching cell types between the tissues. Shown are maximal adjusted p-value for each pairwise randomization: **p<0.01, ***p<0.001. (**C**) PrEDiCT scores of cell types affected by chronic granulomatous disease (CGD) in spleen, lung, and bone marrow. Bold outline marks likely affected cell types. Likely affected cell types that were matching among the tissues were connected by green lines.

The online version of this article includes the following source data and figure supplement(s) for figure 3:

**Source data 1.** The Source Data file contains data used to generate *Figure 3A-C*.

**Figure supplement 1.** Cell-type similarity among human tissues.

**Figure supplement 1—source data 1.** The Source Data file contains data used to generate the circos plot that represents the similarity between cell types of distinct human tissues.

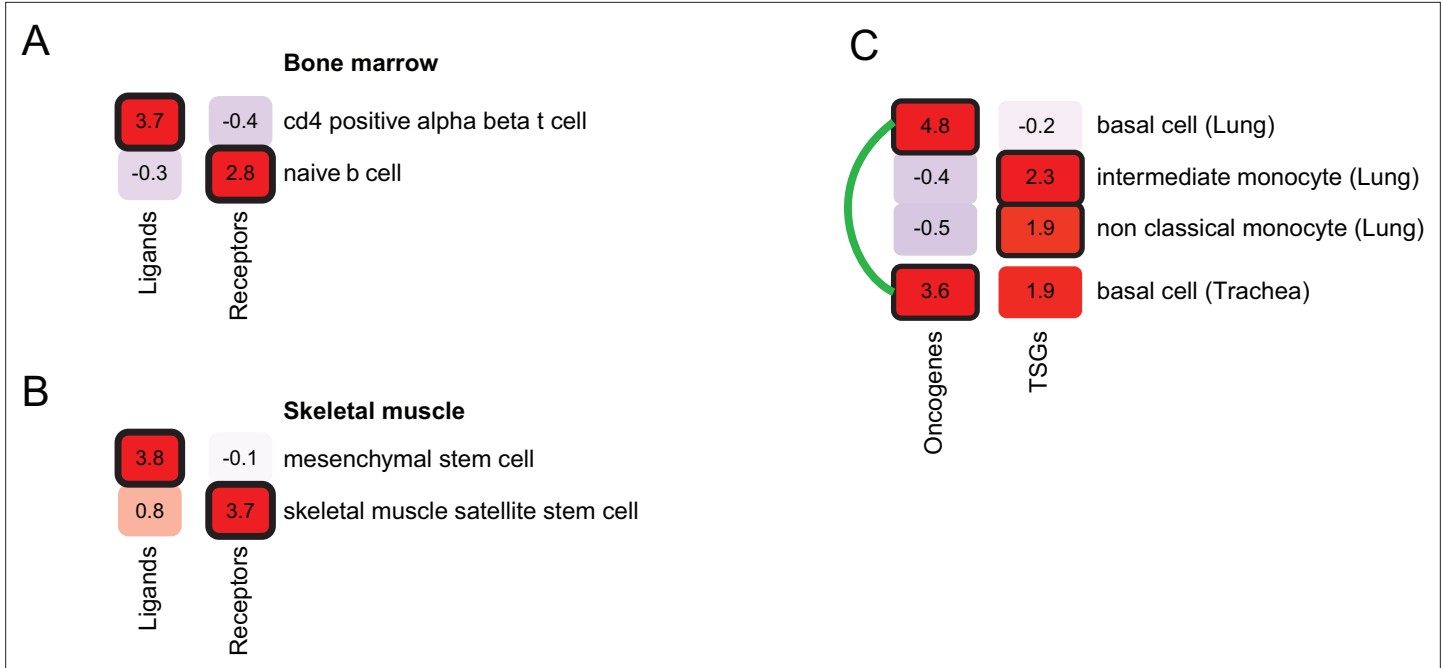

**Figure 4.** Refining cell-type inference using gene functions. (**A**) PrEDiCT scores of cell types likely affected by hyper-IgM immunodeficiency in bone marrow, calculated separately for ligand- or receptor-encoding disease genes. (**B**) PrEDiCT scores of cell types likely affected by autosomal recessive limb-girdle muscular dystrophy in skeletal muscle, calculated separately for ligand- and receptor-encoding disease genes. (**C**) PrEDiCT scores of cell types likely affected by heritable cancers in lung and trachea, calculated separately for oncogenes and tumor suppressor genes. Likely affected cell types that matched between the tissues were connected by a green line. Bold outline marks likely-affected cell types.

The online version of this article includes the following source data for figure 4:

**Source data 1.** The Source Data file contains data used to generate *Figure 4A-C*.

PrEDiCT scheme to LAMA2 (ligand) and DAG1 (receptor), we inferred mesenchymal stem cells and satellite stem cells as the likely affected cell types, respectively (PrEDiCT = 3.8 and 3.7, FDR <0.05; *Figure 4B*). In support of the prediction for DAG1, disruption of DAG1 in satellite stem cells has been associated with the defective muscle regeneration seen in diseased patients (*Cohn et al., 2002*; *Servián-Morilla et al., 2020*). In support of the prediction for LAMA2, the disease was attenuated in ligand-knockout [Col6a1(-/-)] model mice by supplying wild-type mesenchymal-stem-cell derived fibroblasts. This treatment not only restored the collagen VI of the tissue, but also rescued defects in satellite stem cells (*Urciuolo et al., 2013*).

Next, we focused on heritable cancers, where disease genes could be divided by their function to either oncogenes or tumor suppressor genes (TSGs). We retrieved from the Cancer Gene Census (*Sondka et al., 2018*) heritable cancers that manifest in any of the tissues in our dataset (Methods; *Supplementary file 7*). We first applied PrEDiCT scheme jointly to all cancer-associated genes of the same tissue, and then applied it separately to oncogenes or TSGs, which allowed us to infer distinct cell types for oncogenes and TSGs. For example, in airway cancers (lung and trachea), the joint application of PrEDiCT scheme inferred both basal cells and intermediate monocytes as likely affected. The separate application inferred basal cells as likely affected by oncogenes, and intermediate and non-classical monocytes as likely affected by TSGs (*Figure 4C*).

## Characteristics of disease-affected cell types

Are certain cell types more likely to be affected by Mendelian diseases? To answer this, we determined the 'susceptibility' of each cell type as the percentage of diseases affecting it out of all diseases that affect its tissue (*Supplementary file 8*). Most susceptible were capillary endothelial cells of the tongue (Susceptibility = 25%; *Figure 5A*), which were affected by diseases that result in morphologic aberrations of the tongue (e.g., telangiectasia and macroglossia). Next, we asked whether cell-type susceptibility was correlated with cell-type prevalence (i.e., the proportion of its cells out of all cells of

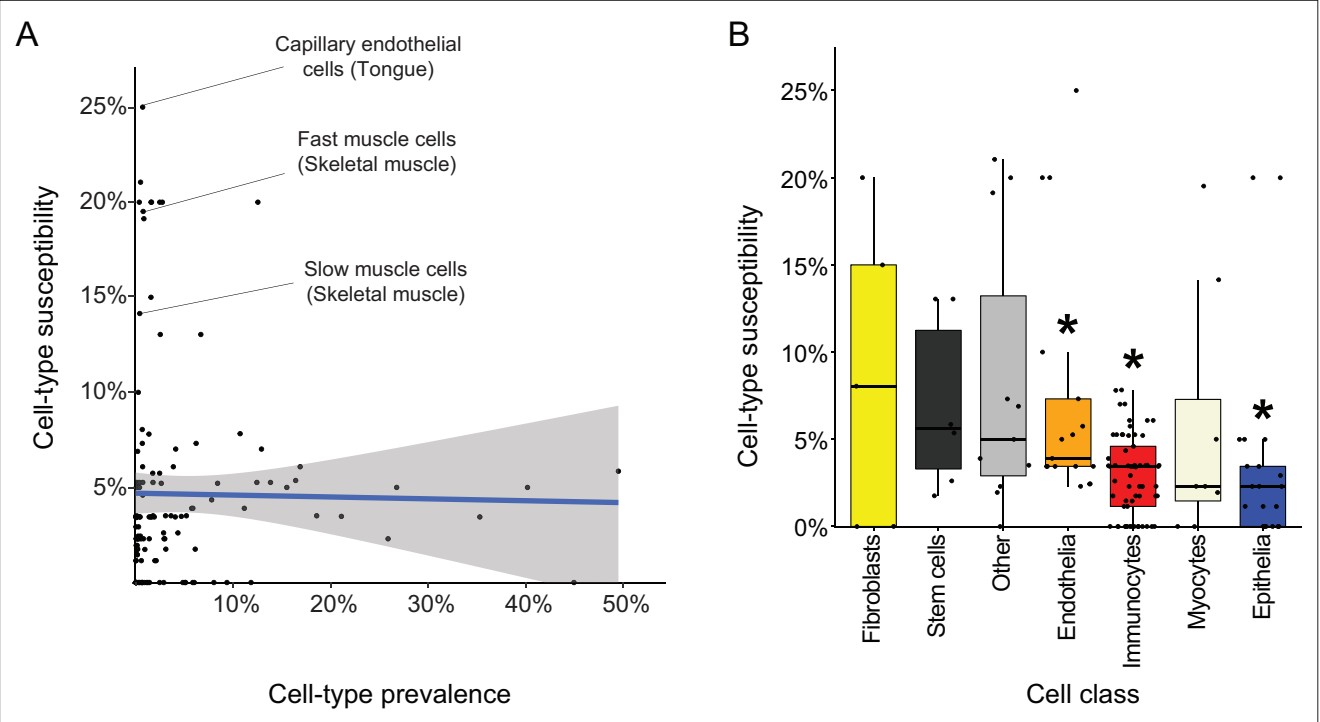

**Figure 5.** Characteristics of disease-affected cell types. (**A**) Cell-type susceptibility (Y-axis) did not correlate with cell-type prevalence (X-axis). The blue line represents linear correlation (r=−0.02, Pearson correlation). (**B**) Cell-type susceptibility varied between cell classes (p=1.5E-3, ANOVA test). Among the cell classes with many cell types shared among tissues, immunocytes and epithelia were the least susceptible, and endothelia were the most susceptible compared to other cell types (adjusted p<0.05, Mann-Whitney U test; Methods).

The online version of this article includes the following source data for figure 5:

**Source data 1.** The Source Data file contains data used to generate *Figure 5A, B*.

its tissue; *Supplementary file 8*). The two measures did not correlate (r=−0.02, Pearson correlation; *Figure 5A*). For example, fast and slow muscle cells were the most susceptible cell types in skeletal muscle, while each accounting for less than 1% of the cells in that tissue.

Lastly, we assessed the susceptibility of cell classes (e.g., capillary endothelial cells were classified as endothelia; *Supplementary file 8*). The most common cell classes were immunocytes, epithelia and endothelia covering 61, 21, and 17 cell types, respectively, in over four tissues. Cell classes had varying susceptibility (*Figure 5B*; p=1.5E-3, ANOVA). Among the common cell classes, immunocytes and epithelia were the least susceptible, and endothelia were the most susceptible compared to other cell types (adjusted p<0.05, Mann-Whitney U test; Methods).

## Discussion

Here, we presented the PrEDiCT scheme for identifying disease-affected cell types based on the cell-type preferential expression of Mendelian disease genes. Preferential expression of disease genes was previously explored in tissue contexts and was shown to characterize disease-affected tissues (*Barshir et al., 2018*; *Barshir et al., 2014*; *Hekselman and Yeger-Lotem, 2020*; *Lage et al., 2008*; *Sonawane et al., 2017*). Recently, cell-type preferential expression has been used to highlight potentially-affected cell types for Mendelian diseases and complex traits, often in combination with cell-type regulatory information and enrichment analyses (*Dai et al., 2021*; *Eraslan et al., 2022*; *Jagadeesh et al., 2021*; *Kim-Hellmuth et al., 2020*; *Zhao et al., 2021*; *Rouhana et al., 2021*; *Zhang et al., 2022*). However, large-scale in-silico validation was rarely conducted (*Montoro et al., 2018*; *Plasschaert et al., 2018*). Here, in contrast, we corroborated likely disease-affected cell types by literature text-mining (*Figure 1*), expert curation (*Figure 1—figure supplement 4*), and recapitulation in mouse (*Figure 2*).

The use of PrEDiCT scheme to identify affected cell types has limitations. Aberrations in disease genes that are preferentially expressed in a cell type do not necessarily lead to disease phenotypes in that cell type, leading to erroneous annotation of disease-affected cell types. For example, metabolic–myopathy-associated genes were upregulated in adipocytes of both muscle and breast, yet only muscle adipocytes showed myopathy phenotypes (*Eraslan et al., 2022*). To reduce the risk for erroneous annotations, we applied PrEDiCT only to cell types of disease-affected tissues (*Figure 1—figure supplement 1*). To enhance the robustness of likely associations, PrEDiCT scores included the cell-type preferential expression of all disease genes, similarly to *Zhang et al., 2022*. For PCD, the PrEDiCT score included 31 genes, and indeed pointed to known disease-affected cell types (*Figure 1C*). Additionally, similarly to other RNA-based schemes, PrEDiCT is oblivious to post-translational regulation, and, since most available single-cell transcriptomic datasets do not contain full-length gene reads, PrEDiCT is also oblivious to cell-type preferential expression of alternatively spliced transcripts. Lastly, preferential expression is just one of several mechanisms that lead to tissue-selective disease manifestations (*Hekselman and Yeger-Lotem, 2020*). Nevertheless, by applying PrEDiCT to 1,140 diseases and single cell transcriptomes of six distinct tissues, we revealed affected cell types for 29% of the Mendelian diseases in our dataset (*Supplementary file 2*). Interestingly, this fits with previous observations that about 30% of Mendelian diseases manifest clinically in a tissue that overexpresses disease genes (*Barshir et al., 2018*; *Barshir et al., 2014*; *Hekselman and Yeger-Lotem, 2020*; *Lage et al., 2008*).

We supported likely disease–cell-type associations by three lines of evidence. The first was text-mining of literature for co-appearance of diseases and cell-types. Text-mining enabled large-scale in-silico assessment, yet co-appearance could also reflect negative and/or speculative results. Our second line of evidence was expert curation. This analysis, although on a smaller scale, provided additional support for the relevance of likely associations (*Figure 1—figure supplement 4*). The third line of evidence came from recapitulation of results using mouse single-cell data. Yet, since the patterns of variation across genes tend to be similar, mouse single-cell data did not provide statistically independent information. This could lead to more false-positive associations for diseases with a single disease gene. However, the fraction of associations that were recapitulated in mouse did not differ between diseases with a single or multiple disease genes, supporting this line of evidence (*Figure 2—figure supplement 1*). Another caveat in the comparison between human and mouse was that gene expression data drove both PrEDiCT calculation and human-mouse cell-type matching. This caveat too had limited impact, since matched cell-types were almost identical upon excluding disease genes. Notably, it would be intriguing to integrate data obtained from human and mouse to increase discovery power in future applications. Altogether, the three lines of evidence provided complementary support for likely associations.

Overall, 328 diseases affected 102/129 cell types. Interestingly, there was no correlation between cell type prevalence and its likelihood to be affected (*Figure 5A*). In particular, endothelia were more likely to be affected by diseases than other prevalent cell classes, whereas immunocytes and epithelia were the least likely to be affected among the prevalent cell classes (*Figure 5B*). This suggests that these cell types are either more resilient than other cell types, or, alternatively, that their impairment is lethal to the organism. To analyze this further, a cross-tissue study of 20 human tissues showed that immunocytes had similar expression signatures across tissues, in accordance with their common functions, whereas endothelia had tissue-specific expression signatures that reflect their tissue-specialized roles (*Jones et al., 2022*). Hence, it seems that germline impairment of immunocytes is more likely lethal, whereas the tissue-specialization of endothelia limits the impact of their germline impairment and thus facilitates overall survival. Yet, our analyses focused on diseases and cell types from six specific tissues and thus were limited in scope. The generalizability of our observations therefore awaits analysis of larger sets of diseases and cell types. In the future, once single cell technologies could offer a comprehensive coverage of expressed genes per cell, it will also be intriguing to assess disease heterogeneity across cells within a cell type.

Our expansive resource of diseases and likely affected cell types could be used to interrogate disease etiologies. For example, we showed that mitochondrial diseases tend to affect muscle cells, in accordance with their energetic demands (*Figure 1E*). Additionally, we showed that diseases inflicting on multiple tissues likely affect similar cell types among those tissues, thereby providing a mechanistic explanation for this phenomenon (*Figure 3*). Lastly, we demonstrated that the inference of likely

affected cell types could be refined by applying PrEDiCT scheme separately to subsets of disease genes with distinct functions. For instance, by separately analyzing oncogenes and TSGs of heritable lung and trachea cancers, we found that oncogenes likely affect basal cells, and TSGs likely affect monocytes (*Figure 4C*). This could suggest that tissue-constructive cells are more susceptible to oncogene mutations, whereas protective cells, such as monocytes, are more susceptible to TSG mutations. The latter is consistent with the function of monocytes in the elimination of malignant transformation of cells in different tissues (*Robinson et al., 2021*). Another interesting subset of diseases are those that impair intercellular communication. A recent study explored the cell types affected by monogenic muscular disorders (*Eraslan et al., 2022*). Consistent with their results, we found that autosomal recessive limb-girdle muscular dystrophy disrupts intercellular communication among muscle cell types, via mutations in the DAG1 receptor (*Figure 4B*). Yet, by applying PrEDiCT separately to the ligand of DAG1, LAMA2, we also highlighted the involvement of mesenchymal stem cells in the disease (*Urciuolo et al., 2013*). By exploring disease genes in appropriate cellular context, we enhanced the mechanistic understanding of disease emergence.

The associations between diseases and affected cell types, though supported by literature and recapitulated in mice, remain putative. Experimental testing could be performed in human cell lines, or in mouse models, in light of the many shared cell types between human and mouse (*Figure 2*). These validation experiments have a huge potential to open new directions in disease research and accelerate cell-directed gene therapy.

## Methods
### Human single-cell transcriptomics analysis
Single-cell transcriptomes were downloaded from Tabula Sapiens (*Jones et al., 2022*). We focused on tissues that consisted of ≥2 samples with ≥800 cells and were sequenced in both human and mouse using microfluidic droplet-based 3'-end technology. These tissues included bone marrow, lung, skeletal muscle, spleen, tongue, and trachea. Analysis was done using Seurat package v4.0.5 (*Hao et al., 2021*). Per tissue, gene expression levels were normalized cell-wise using the NormalizeData function. Henceforth, we considered only genes with normalized counts ≥ 0.05 in ≥10% of cells of at least one cell type (*Supplementary file 9*).

### Mouse single-cell transcriptomics analysis
Single-cell transcriptomes were downloaded from *Tabula Muris, 2018*. To improve the annotation of mouse cells from Tabula Muris, we reanalyzed the transcriptomic profiles of each tissue. First, we selected 2,000 variably expressed genes using FindVariableFeatures function in Seurat. The minimum and maximum average normalized expression of genes across cells were set to 0.05 and 3, respectively (mean.cutoff=c(0.05,3)). We scaled and centered the expression values of the variably expressed genes using ScaleData function, while correcting for the different samples (vars.to.regress='mouse. id'). Then, we projected their expression on all significant principal components (PCs; $p<0.001$, Jack-Straw test) ordered by their explained variance.

Next, we applied a two-phase clustering process. We clustered cells using Seurat FindNeighbors and FindClusters functions based on all the top significant PCs. To resolve over-clustering of cells, we hierarchically ordered cell clusters using BuildClusterTree function. Then, we tested whether cells from different splits in the tree were distinguishable, according to out-of-bag error of a random forest classifier that was trained on variably expressed genes. Indistinguishable cell clusters ($p \geq 0.05$) were merged. To estimate sample-based differences between related clusters, we repeated hierarchically ordering of merged cell clusters. Terminal splits of cell clusters that included uneven numbers of cells from different samples (adjusted $p<0.001$, Chi-square) were merged. Such differences were observed only in tongue and lung tissues. Gene expression normalization and average calculations were applied as described for human tissues. Henceforth, we considered only genes with normalized counts ≥ 0.05 in ≥10% of cells of at least one cell type (*Supplementary file 10*).

### Annotations of mouse cell clusters
We compared the clusters obtained above to the cell type annotations of Tabula Muris. A cluster where a similar annotation was common to >90% of its cells, and where > 90% of cells with that

annotation were within that cluster, was annotated according to Tabula Muris. All other clusters were manually annotated based on highly expressed marker genes (Z score ≥2). We manually searched PubMed for evidence that any of these markers indicates a known cell type (including cell identity and\or function), preferably in the context of the relevant tissue. A cluster was annotated if at least two of its markers indicated the same cell type. To comply with other studies, cell types were named as in Cell Ontology (*Diehl et al., 2016*). Cell type annotations, relevant marker genes and supporting literature appear in *Supplementary file 4*.

### Annotation of diseases-affected tissues

Disease data were retrieved from OMIM (*Amberger et al., 2019*) and included phenotypes with a known molecular basis and phenotypic series. Each disease was associated with its disease genes and their mouse orthologs according to OMIM, and was associated with its affected tissue according to HPO phenotypic abnormality annotations (*Köhler et al., 2021*). We focused on disease that were cataloged by HPO as having main phenotypic abnormality in blood and blood-forming tissues (HP: 0001871), lungs (HP: 0002088), musculature (HP: 0003011), spleen (HP: 0001743), tongue (HP: 0000157), and trachea (HP: 0002778), in accordance with the six tissues that we analyzed. Phenotypic abnormalities that HPO categorized under each of these main terms were also included. We assessed the reliability of associations by comparing them to manually curated associations of diseases and their affected tissues from ODiseA database (*Hekselman et al., 2022*). For this, we downloaded from ODiseA all the diseases affecting blood and bone marrow, lung, and trachea. ODiseA annotations indicating that a tissue is unaffected by a disease were excluded from further analysis.

### PrEDiCT score calculation and significance assessment

For each disease, we calculated its cell-type PrEDiCT scores in all cell types of the disease-affected tissue(s). The calculation was divided into three steps: (i) calculating the cell-type preferential expression of each disease gene; (ii) calculating the cell-type PrEDiCT score; and (iii) assessing the statistical significance of the cell-type PrEDiCT score. The three steps are detailed below.

Step (i): The cell-type preferential expression of a gene was set to the Z-score of its average expression in that cell type relative to cell types of the same tissue (*Equation 1*). Given the sparsity of single-cell data, per tissue, only genes whose average expression in any cell type exceeded the median average expression across genes and cell types were retained. Preferential expression values for human and mouse are available in *Supplementary files 11 and 12*, respectively.

$$P_g^{(c)} = \left[ e_g^{(c)} - average\left(e_g\right) \right] / SD\left(e_g\right) \tag{1}$$

$P_g^{(c)}$ denotes the preferential expression of gene $g$ in cell type $c$. $e_g^{(c)}$ denotes the expression level $e$ of gene $g$ in cell type $c$. Average expression and standard deviation (SD) were calculated across all cell types of an affected tissue.

Step (ii): The PrEDiCT score of disease $D$ in cell type $c$ was set to the median cell-type preferential expression of its disease genes ($g_1$ to $g_n$; *Equation 2*).

$$\text{PrEDiCT}_D^{(c)} = median\left[ P_{g1}^{(c)}, P_{g2}^{(c)} \ldots P_{gn}^{(c)} \right] \tag{2}$$

Step (iii): To assess whether a certain PrEDiCT score was significantly higher than expected by chance for a gene set of the same size, we applied a permutation test. Given a disease $D$ with $n$ disease-associated genes, we selected at random $n$ genes expressed in any cell type, and computed the PrEDiCT score for this random gene set in each cell type of the disease-affected tissue (referred to as 'random score'). We repeated this procedure 1,000 times, resulting in 1,000 random scores per disease and cell type. The p-value of the PrEDiCT score of disease $D$ in cell type $c$ $\left(\text{PrEDiCT}_D^{(c)}\right)$ was set to the fraction of random scores in $c$ that were at least as high as $\text{PrEDiCT}_D^{(c)}$. P-values were then adjusted for multiple hypothesis testing per disease using the Benjamini-Hochberg procedure. The distribution of PrEDiCT scores were similar between tissues (*Supplementary file 13*).

## Text-mining of PubMed records

We searched PubMed for records containing names of the disease in our disease dataset or names of human cell types in the disease-affected tissues. For this, we used eSearch function in Biopython package (*Cock et al., 2009*). The number of maximum records retrieved was set to the maximum that Biopython supports (retmax = 9999). Then, per tissue, we intersected the list of records of each tissue-affecting disease $d$ with the list of records of each tissue cell type $c$, to identify records that mention both. Diseases with less than three records that mentioned it together with a specific cell type were excluded. Cell types that were not mentioned with any disease were excluded. Next, per tissue, we determined whether disease–cell-type pairs significantly co-appeared by applying a Fisher's exact test. Pairs with Z-scores higher than expected (adjusted p<0.001, Bonferroni correction) were determined 'literature-supported'.

## Expert curation and assessment of disease-affected cell types

We assessed whether PrEDiCT score value is indicative of true associations by manual curation of a subset of associations with ranging PrEDiCT score values. Per tissue, we sorted all pairs of diseases and cell-types by their PrEDiCT scores, and then selected two pairs from percentile 0%, 25%, 50%, 75%, and 100%, resulting in a total of 10 pairs per tissue. Each pair was manually reviewed by a medical student to determine whether the cell type presents pathophysiological phenotypes in diseased patients. By using expert knowledge, literature, and OMIM records, each pair was designated as either affected, unaffected, or undetermined.

## Determining matching cell types between tissues

For each pair of tissues, all cell types were compared to each other. For this, each cell type was associated with marker genes, namely genes with Z score ≥2. Markers of mouse cell types were converted to their human orthologous genes according to the Mouse Genome Informatics database (*Bult et al., 2019*). Next, we estimated the similarity between each pair of cell types using matchSCore2 package, which compared the two markers lists by using Jaccard index (*Mereu et al., 2020*). Cell types were determined as matching if their Jaccard index was ≥0.05 and in the top 10th percentile. To test the impact of disease genes on the matching of human and mouse cell types, we repeated the matching upon excluding disease genes from the list of marker genes. Most (109/122, 89%) of the cell-type pairs that matched originally also matched upon excluding disease genes.

## Permutation tests for similarity between likely affected cell types

For each pair of tissues, we denoted one as the reference tissue ($Tr$) and the other as the test tissue ($Tt$). Success was determined, per disease, if any of the cell types in $Tt$ were matching to any of the likely affected cell types in $Tr$. We tested the null hypothesis that the number of diseases for which success was determined ($num\_s$) was not higher than expected by chance. For this, we carried out a permutation test. Per disease, we randomly selected cell types equal in number to the likely affected cell types in $Tr$ and repeated the success test; specifically, we checked if the randomly selected cell types in $Tr$ were matching to any of the likely affected cell types in $Tt$. We repeated this for all diseases with likely affected cell types in $Tr$, and recorded the number of randomly successful diseases ($num\_r$). We repeated this procedure 1000 times. Significance was calculated as the fraction of cases where $num\_r \geq num\_s$. p-Values were adjusted for multiple comparisons by Benjamini-Hochberg correction.

## Analysis of likely affected cell classes

Susceptibility of a given cell type was set to the percentage of diseases affecting that cell type out of all diseases that affect that same tissue (*Supplementary file 7*). Cell types were grouped into one of five cell classes: fibroblasts, stem cells, myocytes, endothelia, epithelia and immunocytes, or were grouped as 'other'. Then, we applied an analysis of variance (ANOVA) test to the proportions of diseases that likely affected each cell class using aov function in R v4.1.1. To further determine whether specific cell classes were more susceptible than others, we compared the cell-type susceptibility of prevalent cell classes (>15 cell types across ≥4 tissues each). to all other cell types, by using Mann-Whitney U test. p-Values were adjusted for multiple comparisons by Benjamini-Hochberg correction.

### Ligands- and receptors-associated diseases

We extracted a list of 1625 pairs of ligands and their corresponding receptors from *Jin et al., 2021*. We retrieved all diseases with disease genes that included both a ligand and its receptor. We filtered this set to include diseases where both the ligand and its receptor were preferentially expressed in distinct cell types.

### Heritable cancers

Data of heritable cancers were downloaded from the Cancer Gene Census of the Catalogue of Somatic Mutations in Cancer [COSMIC; (*Sondka et al., 2018*)]. Specifically, we downloaded germline tumor type, associated genes, and role in cancer (oncogenes or tumor suppressor genes) from tiers 1 and 2. We manually annotated germline tumor types to the six tissues included in our study (*Supplementary file 8*).

## Acknowledgements

This study was funded by the Israel Science Foundation [317/19 to E.Y.-L] and [401/22 to E.Y.-L].

## Additional information

### Funding

| Funder | Grant reference number | Author |
|---|---|---|
| Israel Science Foundation | 317/19 | Esti Yeger-Lotem |
| Israel Science Foundation | 401/22 | Esti Yeger-Lotem |

The funders had no role in study design, data collection and interpretation, or the decision to submit the work for publication.

### Author contributions

Idan Hekselman, Formal analysis, Visualization, Methodology, Writing – original draft; Assaf Vital, Methodology; Maya Ziv-Agam, Lior Kerber, Data curation; Ido Yairi, Data curation, Formal analysis; Esti Yeger-Lotem, Conceptualization, Supervision, Writing – original draft

### Author ORCIDs

Idan Hekselman http://orcid.org/0000-0003-0529-7838
Lior Kerber http://orcid.org/0009-0005-0583-5618
Esti Yeger-Lotem https://orcid.org/0000-0002-8279-7898

### Decision letter and Author response

Decision letter https://doi.org/10.7554/eLife.84613.sa1
Author response https://doi.org/10.7554/eLife.84613.sa2

## Additional files

### Supplementary files

- Supplementary file 1. Diseases, disease genes, and likely affected cell types of each tissue.
- Supplementary file 2. Diseases and their PrEDiCT scores across human cell types per tissue.
- Supplementary file 3. Names of disease and cell type co-appearance in PubMed records per tissue.
- Supplementary file 4. Mouse cell clusters annotations.
- Supplementary file 5. Matching cell types between human and mouse tissues.
- Supplementary file 6. Diseases and their PrEDiCT scores across mouse cell types per tissue.
- Supplementary file 7. Tissues affected by heritable cancers.
- Supplementary file 8. Prevalence, susceptibility and classes of cell types.

- Supplementary file 9. The percentage of cells that express a gene per cell type in human tissues.
- Supplementary file 10. The percentage of cells that express a gene per cell type in mouse tissues.
- Supplementary file 11. The preferential expression of genes in cell types of human tissues.
- Supplementary file 12. The preferential expression of genes in cell types of mouse tissues.
- Supplementary file 13. Summary of PrEDiCT scores.
- MDAR checklist

## Data availability

All data generated or analyzed during this study are included in the manuscript and in the supporting files. The Source Data files contain data used to generate all figures. Additional data and code to redo analysis are available at GitHub https://github.com/hekselman/PrEDiCT(copy archived at *Hekselman, 2024*) and Dryad https://doi.org/10.5061/dryad.9w0vt4bm7.

The following dataset was generated:

| Author(s) | Year | Dataset title | Dataset URL | Database and Identifier |
|---|---|---|---|---|
| Hekselman I, Yeger-Lotem E | 2024 | Affected cell types for hundreds of Mendelian diseases revealed by analysis of human and mouse single-cell data | https://doi.org/10.5061/dryad.9w0vt4bm7 | Dryad, 10.5061/dryad.9w0vt4bm7 |

The following previously published datasets were used:

| Author(s) | Year | Dataset title | Dataset URL | Database and Identifier |
|---|---|---|---|---|
| The Tabula Sapiens Consortium, Jones RC, Karkanias J, Krasnow MA | 2022 | Tabula Sapiens | https://www.ncbi.nlm.nih.gov/geo/query/acc.cgi?acc=GSE201333 | NCBI Gene Expression Omnibus, GSE201333 |
| Tabula Muris Consortium | 2018 | Tabula Muris: Transcriptomic characterization of 20 organs and tissues from *Mus musculus* at single cell resolution | https://www.ncbi.nlm.nih.gov/geo/query/acc.cgi?acc=GSE109774 | NCBI Gene Expression Omnibus, GSE109774 |

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
