## [Editor Report]

The study presents analyses linking cell-types to monogenic disorders using over-expression of known disease-associated genes in single-cell data to identify disease-affected cell types for 328 Mendelian diseases. Overall, this important study combines multiple data analyses to quantify the connection between cell types and human disorders. Compelling analyses using stringent and rigorous statistical methodologies support the conclusions of this study.

---

## [Decision Letter]

**Decision letter after peer review:**

Thank you for submitting your article "Affected cell types for hundreds of Mendelian diseases revealed by analysis of human and mouse single-cell data" for consideration by *eLife*. Your article has been reviewed by 3 peer reviewers, one of whom is a member of our Board of Reviewing Editors, and the evaluation has been overseen by Molly Przeworski as the Senior Editor. The reviewers have opted to remain anonymous.

Essential revisions:

1. All reviewers commented with respect to lack of rigor in the approach for determining statistical significance (Rev1#1, Rev2#1, Rev3#2). Criteria for statistical significance needs to be justified/investigated to provide statistical stringency for determining the set of significant associations.

2. Investigation of null/negative control genes would strengthen the conclusions of this work (Rev3#1, Rev2#4). For example, some Mendelian diseases impact disease through coding mutations without impacting expression in any cell type. Such genes could serve as negative controls showcasing the method does not identify any disease-affected cell type.

3. Improved textual contextualization (and caveats) of proposed work to better showcase the proposed results would significantly strengthen the rigor (Rev1#2, Rev3#1,4,5,6).

*Reviewer #1 (Recommendations for the authors):*

The manuscript by Hekselman et al. presents analyses linking cell-types to monogenic disorders using over-expression of monogenic disease genes as the signal. The manuscript analyses data from 6 tissues (bone marrow, lung, muscle, spleen, tongue and trachea) together with ~1,000 rare diseases from OMIM (with ~2,000 associated genes) to identify cell-type of interest for specific disease of choice. The signal used by the approach is the relative expression of OMIM-genes in a particular cell type relative to the expression of the gene in the tissue of interest identifying cell-type-disease pairs that are then investigated through literature review and recapitulated using mouse expression. A potentially interesting finding is that disease genes manifesting in multiple tissues seem to hit same cell-types. Overall, this important study combines multiple data analyses to quantify the connection between cell types and human disorders. However, whereas some of the analyses are compelling, the statistical analyses are incomplete as they don't provide full treatment of type I error.

I have two main critiques that reduce my enthusiasm for this work.

1. The statistical framework to identify cell-types for a given disease (gene) is likely not calibrated to type I error (page 5 and methods). The approach notes that disease-cell-type pairs with a PREDICT score > 2 (top 5%) of all pairs are "considered significant". I could not find any justification for why this is a well-calibrated test that yields significant associations. Under a model of where no true cell-type-associations are present in the data, this approach will still incorrectly flag the top 5% as being significant. In fact, Figure 1B appears consistent with PREDICT scores drawn from a normal distribution centered on 0 potentially consistent with most (if not all) associations being null. The approach needs a proper statistical model that is well-calibrated test to declare significant associations.

2. Functional validation of the disease-cell-type is limited. Associations are investigate using literature search in Pubmed focused on co-occurrence of disease and cell types; this is not validation as many of those co-appearances reflect negative and/or speculative results. The second line of evidence that these associations are true comes from recapitulation of results using mouse data; these results could be over-interpreted as human and mouse cell type data is matched based on expression itself thus creating an expected loop: mouse cell-type expression matched to human cell-type expression identifies similar expression-driven associations.

*Reviewer #2 (Recommendations for the authors):*

Comments:

1. The PrEDiCT score threshold for inferring a disease-cell type pair as statistically significant seems unlikely to be sufficiently stringent. The threshold for inferring statistical significance must be rigorously justified.

First, for diseases with only 1 associated gene, z>2 is a lax threshold, as 2.5% of disease-cell type pairs would be expected to be significant by chance. The distribution of the number of associated genes per disease is important and must be discussed in the main text with an associated Supp Table/Supp Figure (how many of the 1113 diseases have only 1 associated gene, how many have exactly 2 associated genes, etc.).

Second, from Table S3, it seems that the total number of disease-cell type pairs tested is 3952 + 5624 + 12996 + 2568 + 780 + 540 = 26460 (which is less than 1113*129 = 143577, because only cell types in disease-associated tissues were tested). The number is 26460 is important and must be reported in the main text. Given that 2.5% * 26460 = 662, in the extreme case that all diseases have only 1 associated gene we would expect 662 significant PrEDiCT scores by chance, such that many of the 753 significant PrEDiCT scores that are reported could be false positives.

Third, under the null hypothesis of no disease-affected cell types, for diseases with only 1 associated gene, the PrEDiCT scores may have more large values than a normal distribution, because there could be many genes with patterns of preferential expression in a particular cell type. On the other hand, for diseases with x associated genes with x>1, the PrEDiCT scores may have fewer large values than a normal distribution, because the median of x independent z-scores has a <2.5% chance of being >2.

The best solution would be to assess statistical significance via empirical comparison with PrEDiCT scores for non-disease-associated control genes (for diseases with only 1 associated gene), or empirical comparison with PrEDiCT scores for the median of x non-disease-associated control genes (for diseases with x associated genes with x>1). I recommend that this approach should be used. The resulting P-values can then be evaluated for statistical significance using Bonferroni (probably too conservative) or FDR. It is likely that the method has higher power for larger values of x, such that FDR could be stratified by the value of x. Note that the main point here is variation across genes: unless disease-associated gene(s) are very significantly different from non-disease-associated control genes, then no significant disease-affected cell type should be inferred.

An alternative, which I have much less enthusiasm for but which is plausible, would be to state throughout that the PrEDiCT scores identify *candidate* disease-affected cell types, and use excess overlap with disease-cell type pairs from literature co-appearance to assign an FDR to the *candidate* disease-affected cell types (perhaps at different PrEDiCT score thresholds), e.g., FDR = 100% / (fold excess overlap).

2. The numbers in Figure 1E are not consistent with the numbers in the main text.

First, the main text (p.5) states that 753 disease-cell type pairs have significant literature co-appearance, whereas Figure 1E states that 654+99=753 disease-cell type pairs have significant PrEDiCT scores and 448+99=547 pairs have significant literature co-appearance. I am guessing that 753 in the main text is a typo and should be 547.

Second, the main text (p.6) states that 714 diseases had disease-affected cell types inferred (I believe this is based on the PrEDiCT score, but the wording of the text is confusing and could be improved), whereas Figure 1E states that 654+99=753 disease-cell type pairs have significant PrEDiCT scores. I’m guessing that 753>714 because some diseases have >1 disease-affected cell types inferred, but this should be stated explicitly.

Third, the main text (p.6) states that 18% of disease-cell type pairs with significant PrEDiCT scores have significant literature co-appearance. However, based on the numbers in Figure 1E, 99/753 = 13%, which is different from 18%.

Fourth, the main text (p.6) states that 6% of all disease-cell type pairs have significant literature co-appearance. However, based on 547 from Figure 1E and the total of 26460 disease-cell type pairs tested (see Comment 1), 547/26460 = 2%, which is different from 6%.

In addition to fixing the discrepancies, it may be good to expand the explanations, as this is really the most important part of the paper.

3. The PCD example (Figure 1C, Figure 2D) is a great example. However, I suggest to increase the number of specific Mendelian diseases highlighted in the main Figures (prior to delving into diseases with multiple affected tissues, or other special categories of diseases) from 1 to at least 4, e.g., via a separate main Figure highlighting 4 Mendelian diseases (with human or human+mouse results).

4. I expect that for some Mendelian diseases, coding mutations to disease-associated gene(s) affect protein product but do not affect expression in any cell type. It would be interesting to include some well-studied Mendelian diseases in this category as negative controls, for which failure to implicate any disease-affected cell type is the correct answer.

*Reviewer #3 (Recommendations for the authors):*

I really do believe the authors should provide context of their method with a larger set of previously published work. Additionally, the thresholding concerns and lack of a clear null comparison make it difficult to assess the robustness of the method and the analyses.

---

## [Author Response]

Essential revisions:1. All reviewers commented with respect to lack of rigor in the approach for determining statistical significance (Rev1#1, Rev2#1, Rev3#2). Criteria for statistical significance needs to be justified/investigated to provide statistical stringency for determining the set of significant associations.

We changed the approach for determining the statistical significance of PrEDiCT scores to be based on permutation tests, as suggested by the reviewers (Results, page 6, 1^st^ paragraph; Methods, page 21-22, ‘PrEDiCT score calculation and significance assessment’; Figure 1—figure supplement 2). We assessed the stringency of the approach using literature text-mining and expert curation (Results, page 7, 2^nd^ paragraph; Methods, page 22, ‘Text-mining of PubMed records’, and page 23, ‘Expert curation and assessment of disease-affected cell types’; Figure 1F,G, and Figure 1—figure supplement 4).

2. Investigation of null/negative control genes would strengthen the conclusions of this work (Rev3#1, Rev2#4). For example, some Mendelian diseases impact disease through coding mutations without impacting expression in any cell type. Such genes could serve as negative controls showcasing the method does not identify any disease-affected cell type.

We added negative control cases to the Results, page 7, 1^st^ paragraph.

3. Improved textual contextualization (and caveats) of proposed work to better showcase the proposed results would significantly strengthen the rigor (Rev1#2, Rev3#1,4,5,6).

We improved textual contextualization in the Introduction (from the end of page 2 to page 3) and throughout the manuscript. We extended the discussion of caveats in the Discussion (page 16, 2^nd^ paragraph and page 17, 2^nd^ paragraph).

Reviewer #1 (Recommendations for the authors):The manuscript by Hekselman et al. presents analyses linking cell-types to monogenic disorders using over-expression of monogenic disease genes as the signal. The manuscript analyses data from 6 tissues (bone marrow, lung, muscle, spleen, tongue and trachea) together with ~1,000 rare diseases from OMIM (with ~2,000 associated genes) to identify cell-type of interest for specific disease of choice. The signal used by the approach is the relative expression of OMIM-genes in a particular cell type relative to the expression of the gene in the tissue of interest identifying cell-type-disease pairs that are then investigated through literature review and recapitulated using mouse expression. A potentially interesting finding is that disease genes manifesting in multiple tissues seem to hit same cell-types. Overall, this important study combines multiple data analyses to quantify the connection between cell types and human disorders. However, whereas some of the analyses are compelling, the statistical analyses are incomplete as they don't provide full treatment of type I error.I have two main critiques that reduce my enthusiasm for this work.1. The statistical framework to identify cell-types for a given disease (gene) is likely not calibrated to type I error (page 5 and methods). The approach notes that disease-cell-type pairs with a PREDICT score > 2 (top 5%) of all pairs are "considered significant". I could not find any justification for why this is a well-calibrated test that yields significant associations. Under a model of where no true cell-type-associations are present in the data, this approach will still incorrectly flag the top 5% as being significant. In fact, Figure 1B appears consistent with PREDICT scores drawn from a normal distribution centered on 0 potentially consistent with most (if not all) associations being null. The approach needs a proper statistical model that is well-calibrated test to declare significant associations.

Following this comment, we revised the procedure for declaring significant disease–cell-type associations. Instead of considering disease–cell-type pairs with PrEDiCT score > 2 (top 5%) as significant, we used permutation testing to determine statistical significance Results, page 6, 1^st^ paragraph; Methods, page 21-22, ‘PrEDiCT score calculation and significance assessment’; (Figure 1—figure supplement 2). Specifically, given a disease D with n disease-associated genes, we randomly selected n genes expressed in any cell type, and computed the PrEDiCT score for this random gene set in each cell type of the disease-affected tissue (referred to as ‘random score’). We repeated this procedure 1,000 times, resulting in 1,000 random scores per disease and cell type. The p-value of the PrEDiCT score of disease D in cell type c was set to the fraction of random scores in c that were at least as high as the original PrEDiCT score of D in c. P-values were adjusted for multiple hypothesis testing per disease using the Benjamini-Hochberg procedure. To increase stringency, we treated only statistically significant disease–cell-type pairs with PrEDiCT score≥1 as 'likely affected'. We estimated type I error using literature text-mining or expert curation (Figure 1F and Figure 1—figure supplement 4A). False-positive rates of the revised procedure were low in both (0.01 and 0.07, respectively).

2. Functional validation of the disease-cell-type is limited. Associations are investigate using literature search in Pubmed focused on co-occurrence of disease and cell types; this is not validation as many of those co-appearances reflect negative and/or speculative results. The second line of evidence that these associations are true comes from recapitulation of results using mouse data; these results could be over-interpreted as human and mouse cell type data is matched based on expression itself thus creating an expected loop: mouse cell-type expression matched to human cell-type expression identifies similar expression-driven associations.

We agree with the reviewer that functional validation using literature search in Pubmed is limited. To strengthen the reliability of likely disease–cell-type associations, we assigned an expert to curate associations with different PrEDiCT scores (ten associations per tissue, for six tissues; Methods, page 23, ‘Expert curation and assessment of disease-affected cell types’). Next, we used the curated association to estimate false-positive and false-negative rates. This analysis showed that the revised scheme had a low false-positive rate and that likely associations were enriched in expert-verified pairs (p=1.3E-4, Fisher’s exact test; Figure 1—figure supplement 4).

With respect to the recapitulation of results using mouse data, we assessed the impact of 'reuse' of expression data to drive associations. To avoid reusing disease genes in both PrEDiCT scheme and in the matching between human and mouse cell types, we matched between cell types based on the expression of non-disease genes alone. We found that most of the originally matched celltype pairs (109/122, 89%) were also matched upon excluding disease genes. Hence, the impact of 'reuse' of expression data seems minor. We acknowledge this caveat and mention the observation described herein in Discussion, page 17, 2^nd^ half of the 2^nd^ paragraph; Results, page 8, last sentence; and Methods, page 23, ‘Determining matching cell types between tissues’.

Reviewer #2 (Recommendations for the authors):Comments:1. The PrEDiCT score threshold for inferring a disease-cell type pair as statistically significant seems unlikely to be sufficiently stringent. The threshold for inferring statistical significance must be rigorously justified.First, for diseases with only 1 associated gene, z>2 is a lax threshold, as 2.5% of disease-cell type pairs would be expected to be significant by chance. The distribution of the number of associated genes per disease is important and must be discussed in the main text with an associated Supp Table/Supp Figure (how many of the 1113 diseases have only 1 associated gene, how many have exactly 2 associated genes, etc.).Second, from Table S3, it seems that the total number of disease-cell type pairs tested is 3952 + 5624 + 12996 + 2568 + 780 + 540 = 26460 (which is less than 1113*129 = 143577, because only cell types in disease-associated tissues were tested). The number is 26460 is important and must be reported in the main text. Given that 2.5% * 26460 = 662, in the extreme case that all diseases have only 1 associated gene we would expect 662 significant PrEDiCT scores by chance, such that many of the 753 significant PrEDiCT scores that are reported could be false positives.Third, under the null hypothesis of no disease-affected cell types, for diseases with only 1 associated gene, the PrEDiCT scores may have more large values than a normal distribution, because there could be many genes with patterns of preferential expression in a particular cell type. On the other hand, for diseases with x associated genes with x>1, the PrEDiCT scores may have fewer large values than a normal distribution, because the median of x independent z-scores has a <2.5% chance of being >2.The best solution would be to assess statistical significance via empirical comparison with PrEDiCT scores for non-disease-associated control genes (for diseases with only 1 associated gene), or empirical comparison with PrEDiCT scores for the median of x non-disease-associated control genes (for diseases with x associated genes with x>1). I recommend that this approach should be used. The resulting P-values can then be evaluated for statistical significance using Bonferroni (probably too conservative) or FDR. It is likely that the method has higher power for larger values of x, such that FDR could be stratified by the value of x. Note that the main point here is variation across genes: unless disease-associated gene(s) are very significantly different from non-disease-associated control genes, then no significant disease-affected cell type should be inferred.An alternative, which I have much less enthusiasm for but which is plausible, would be to state throughout that the PrEDiCT scores identify candidate disease-affected cell types, and use excess overlap with disease-cell type pairs from literature co-appearance to assign an FDR to the candidate disease-affected cell types (perhaps at different PrEDiCT score thresholds), e.g., FDR = 100% / (fold excess overlap).

We thank the reviewer for these suggestions and adopted the ‘best solution’ that the reviewer suggested above. Specifically, we empirically compared the original cell-type PrEDiCT scores of a disease with PrEDiCT scores computed for randomly selected sets of non-disease-associated control genes. The resulting P-values were then adjusted using Benjamini-Hochberg procedure (Results, page 6, 1^st^ paragraph; Methods, page 21-22, ‘PrEDiCT score calculation and significance assessment’; Figure 1—figure supplement 2).

Per the reviewer’s requests, we added a supplementary figure that shows the distribution of diseases by the number of associated genes, and the same distribution for the diseases with likely associations (Figure 1—figure supplement 3). We report the revised total number of disease–celltype pairs tested in Results, page 6, end of 1^st^ paragraph.

2. The numbers in Figure 1E are not consistent with the numbers in the main text.First, the main text (p.5) states that 753 disease-cell type pairs have significant literature co-appearance, whereas Figure 1E states that 654+99=753 disease-cell type pairs have significant PrEDiCT scores and 448+99=547 pairs have significant literature co-appearance. I am guessing that 753 in the main text is a typo and should be 547.Second, the main text (p.6) states that 714 diseases had disease-affected cell types inferred (I believe this is based on the PrEDiCT score, but the wording of the text is confusing and could be improved), whereas Figure 1E states that 654+99=753 disease-cell type pairs have significant PrEDiCT scores. I’m guessing that 753>714 because some diseases have >1 disease-affected cell types inferred, but this should be stated explicitly.Third, the main text (p.6) states that 18% of disease-cell type pairs with significant PrEDiCT scores have significant literature co-appearance. However, based on the numbers in Figure 1E, 99/753 = 13%, which is different from 18%.Fourth, the main text (p.6) states that 6% of all disease-cell type pairs have significant literature co-appearance. However, based on 547 from Figure 1E and the total of 26460 disease-cell type pairs tested (see Comment 1), 547/26460 = 2%, which is different from 6%.In addition to fixing the discrepancies, it may be good to expand the explanations, as this is really the most important part of the paper.

We are grateful to the reviewer for noting these issues. We recalculated the various numbers after redoing the entire analysis, including revising the disease-tissue dataset, calculation of PrEDiCT scores and their statistical significance, and rerunning the Biopython package to identify disease–cell-type pairs with significant literature coappearance. Consequently, the results have changed, and we revised all relevant numbers (also summarized in new Supplementary File 1A). For the first and second point, the revised number of disease–cell-type pairs with significant PrEDiCT scores, denoted 'likely affected associations', was 489. The disease–cell-type pairs with significant literature co-appearance was revised to 229 pairs (denoted 'literature-supported pairs'). Importantly, the current version of Biopython retrieves only up to 9,999 records per term, rather than the previously available threshold of 99,999 records. For instance, Biopython retrieved 9,999 records for each of the terms erythrocytes and β-thalassemia, of which 32 records overlapped (the overlap remained statistically significant). We revised wording to explicitly state the revised numbers in Results, page 7, last sentence; and in Figure 1G (also see figure legend).

For the third and fourth points, we recalculated the different fractions. The fraction of likely affected disease–cell-type pairs that are literature-supported out of all likely affected pairs was updated to 41/489 (8.4%). The fraction of literature-supported disease–cell-type pairs out of all disease–celltype pairs was updated to 229/34,249 (0.7%, now in Supplementary File 2A; the number of all disease–cell-type pairs grew due to miscalculation in the original analysis). We updated the text and expanded the explanations in Results, page 8, 1^st^ paragraph; and in Figure 1G.

3. The PCD example (Figure 1C, Figure 2D) is a great example. However, I suggest to increase the number of specific Mendelian diseases highlighted in the main Figures (prior to delving into diseases with multiple affected tissues, or other special categories of diseases) from 1 to at least 4, e.g., via a separate main Figure highlighting 4 Mendelian diseases (with human or human+mouse results).

Per the reviewer’s suggestion, we highlight five additional Mendelian diseases (Results, page 7, 1^st^ paragraph; Figure 1D and 1E), all of which were recapitulated using mouse data (Results, page 9, 1^st^ paragraph; Figure 2D and 2E).

4. I expect that for some Mendelian diseases, coding mutations to disease-associated gene(s) affect protein product but do not affect expression in any cell type. It would be interesting to include some well-studied Mendelian diseases in this category as negative controls, for which failure to implicate any disease-affected cell type is the correct answer.

We thank the reviewer for this suggestion. We included two cases of negative controls, for which failure to implicate any disease-affected cell type is the correct answer in Results, page 7, 1^st^ paragraph.

Reviewer #3 (Recommendations for the authors):I really do believe the authors should provide context of their method with a larger set of previously published work. Additionally, the thresholding concerns and lack of a clear null comparison make it difficult to assess the robustness of the method and the analyses.

We provide context for our method by providing a detailed description of a larger set of previously published work (Introduction, page 2-3). Additionally, we revised the statistical analysis and clarified the null comparison of the results (Results, page 6, 1^st^ paragraph; Methods, page 21-22, ‘PrEDiCT score calculation and significance assessment’; Figure 1—figure supplement 2). We enhanced the assessments of the method performance and robustness by using several lines of external evidence (Results, page 7, 2^nd^ paragraph; Methods, page 22, ‘Text-mining of PubMed records’, and page 23, ‘Expert curation and assessment of disease-affected cell types’; Figure 1F,G, and Figure 1—figure supplement 4).